# Climate-driven succession in marine microbiome biodiversity and biogeochemical function

Alyse A. Larkin [1], Melissa L. Brock [2], Adam J. Fagan [1], Allison R. Moreno [2,3], Skylar D. Gerace [1], Lauren E. Lees [2], Stacy A. Suarez[2], Emiley A. Eloe-Fadrosh [4] & Adam C. Martiny [1,2] ✉

Seasonal and El Niño-Southern Oscillation (ENSO) warming result in similar ocean changes as predicted with climate change. Climate-driven environmental cycles have strong impacts on microbiome diversity, but impacts on microbiome function are poorly understood. Here we quantify changes in microbial genomic diversity and functioning over 11 years covering seasonal and ENSO cycles at a coastal site in the southern California Current. We observe seasonal oscillations between large-genome lineages during cold, nutrient rich conditions in winter and spring versus small-genome lineages, including *Prochlorococcus* and *Pelagibacter*, in summer and fall. Parallel interannual changes separate communities depending on ENSO condition. Biodiversity shifts translate into clear oscillations in microbiome functional potential. Ocean warming induced an ecosystem with less iron but more macronutrient stress genes, depressed organic carbon degradation potential and biomass, and elevated carbon-to-nutrient biomass ratios. The consistent microbial response observed across time-scales points towards large climate-driven changes in marine ecosystems and biogeochemical cycles.

Empirical observations of microbial responses to marine warming are necessary to better understand nonlinear, climate-driven biophysical interactions between the environment and plankton communities. Previous work has demonstrated that warming events, including marine heatwaves, can have profound effects on the biodiversity of microbial communities[1,2]. For example, a 2015–16 marine heatwave in the Tasman Sea resulted in a shift in microbial communities to niche states that resembled locations over 1000 km equatorward[3]. However, climate-driven impacts on microbial functional potential are poorly quantified. For example, only a limited number of studies have examined metagenomic functional potential for periods longer than 5 years[4–6]. Moreover, differing hypotheses exist as to whether taxonomic changes in microbial communities result in changing ecosystem processes[7–9]. Some studies demonstrate widespread functional redundancy[10], while others have observed global variability in functional potential for traits including nutrient uptake[11], predicted growth and replication[12,13], and carbon metabolism[14]. Thus, whether temperature-driven changes in marine microbial communities may impact ecosystem functional potential remains an outstanding question.

El Niño-driven warming events result in many of the same environmental changes as predicted with climate change. However, observations of El Niño's impact on bacterioplankton communities are limited. The 2015 El Niño event was one of the strongest on record in terms of the spatial extent of warming in the Eastern North Pacific[15]. In the California Current, phytoplankton communities shifted to increased cyanobacteria cell abundances[16] and zooplankton communities showed significant changes in composition[17]. In addition,

[1]Department of Earth System Science, University of California, Irvine, CA, USA. [2]Department of Ecology and Evolutionary Biology, University of California, Irvine, CA, USA. [3]Ocean Sciences Department, University of California, Santa Cruz, CA, USA. [4]US Department of Energy Joint Genome Institute, Lawrence Berkeley National Laboratory, Berkeley, CA, USA. ✉e-mail: amartiny@uci.edu

common microbial taxa including SAR11, *Synechococcus*, and *Prochlorococcus* shifted from cold to warm water ecotypes[18,19]. In terms of biogeochemical impact, the 2015 El Niño suppressed primary production[20], biomass, and particulate organic matter concentrations, and increased the carbon-to-nutrient ratios[21]. Given the significant impacts on microbial community composition and marine biogeochemistry, the 2015 El Niño offers an opportunity to examine the impact of warming temperatures on microbial metabolism as well as establish links between microbial community composition, functional potential, and ecosystem impacts.

Within the California Current Ecosystem, the Southern California Bight (SCB) is a critical transition zone between subpolar and subtropical biomes. El Niño-Southern Oscillation (ENSO) cycles affect the region directly through warming and also indirectly through circulation changes. This region is influenced by the southward, cold, nutrient poor, and low salinity California Current as well as the northward, warm, nutrient rich, and high salinity California Undercurrent[22]. In addition, the SCB is strongly impacted by both point and non-point source pollution, which can induce high levels of localized nutrient loading[23]. Past studies have found clear seasonal cycles in environmental conditions and microbial taxonomic diversity both near- and offshore[21,24,25]. Thus, this region represents an excellent model system to test the impact of diverse environmental changes on marine microbiomes.

Here, we combine environmental analyses and metagenomic sequencing from 2011 to 2022 to quantify links between in situ warming, changing nutrient inputs, microbial biodiversity, and ecosystem functions. In particular, we identify the major temporal modes of variation in microbial biodiversity and community-level genomic functional potential. We hypothesize clear impacts of seasonal and ENSO cycles (including the strong 2015 El Niño event). Thus, we predict that seasonal and El Niño-driven warming will result in parallel increases of oligotrophic bacterioplankton taxa with a concurrent increase in metabolic strategies associated with oligotrophic communities. We aim to establish a genomics-enabled understanding of

the climate-driven feedbacks between ocean environmental change, microbial diversity, and biogeochemistry.

## Results

To identify the drivers of seasonal and interannual succession of marine microbial communities, we quantified the environmental conditions using a combination of discrete water samples (nitrate, phosphate, and particulate organic matter) and in situ sensors (temperature and chlorophyll). Samples were taken at the MiCRO site located near-shore in the SCB. In addition, we collected 267 metagenomes covering 11 years (2011–2021) (Supplementary Data 1). These samples spanned all months as well as several stages of the ENSO cycle (Fig. 1A). A total of 3.47 Tbp were sequenced across all samples.

### Environmental dynamics

We observed clear seasonal oscillations in temperature and nutrients (Fig. 1). As usual for this latitude (33˚N), sea-surface temperature peaked between July and September. Nutrients mostly showed an inverse oscillation with high concentrations during the winter and early spring when upwelling occurs[26]. Coinciding with phytoplankton net growth (reflected in rising chlorophyll concentrations), there was a nutrient decline between March and June resulting in low concentrations during the summer and fall. However, nitrate and phosphate showed unique trends in the fall, whereby phosphate rose but nitrate stayed at low levels leading to a shift in the dissolved nitrate:phosphate. The ENSO climate cycle also had an imprint on the environmental conditions. Interannually, temperature was depressed during La Niña conditions from 2011 to 2013, high during the strong El Niño event in 2014–2016, a period of intermediate conditions, and then another La Niña event in 2021. Nutrients generally inversely followed temperature trends with higher levels during La Niña and lower during El Niño conditions. However, phosphate was elevated in 2017 and 2018 despite above average temperature. Thus, we observed a clear correspondence between temperature and nutrients at both seasonal and interannual time-scales but also variation in the relationship between temperature and specific nutrients.

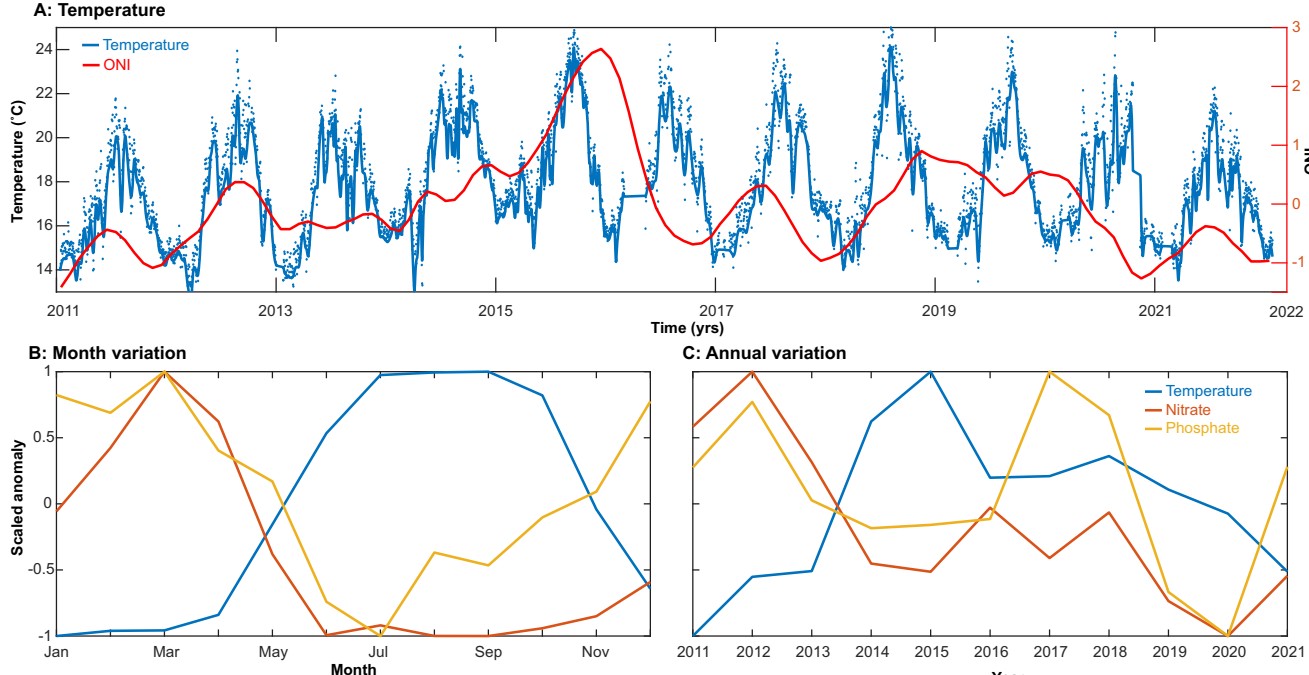

**Fig. 1 | Environmental dynamics across multi time-scales. A** Temperature (°C) measured at the SCCOOS automated shore-station and the Ocean Nino Index (ONI). **B** Monthly anomaly in temperature, nitrate and phosphate. **C** 2011 to 2021 annual anomalies in temperature, nitrate and phosphate. Anomalies were found by fitting linear models with 12 monthly and yearly levels and scaled from −1 to 1 for presentation. The full time-series of nitrate and phosphate are presented in Supplementary Fig. 1.

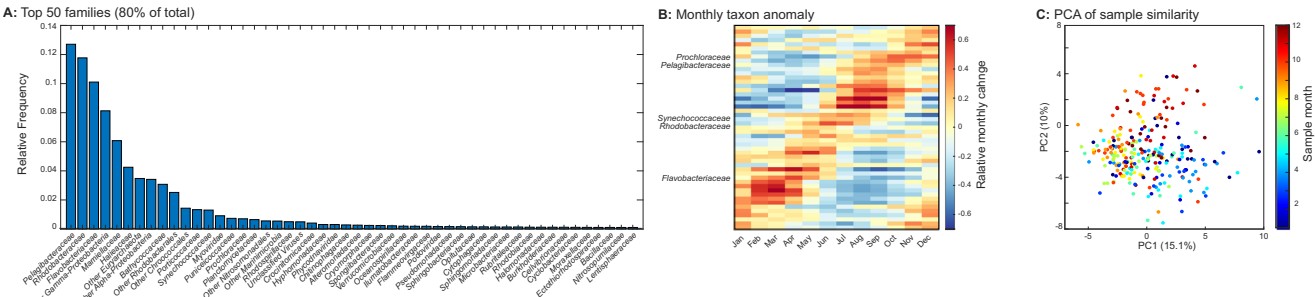

**Fig. 2 | Dominance and seasonal dynamics of key lineages. A** Rank abundance of top 50 lineages. The top 50 lineages (defined at the family level) represented 80% of all sequences classified ($n = 929$). **B** Seasonal succession of top 50 lineages. A linear model with monthly and yearly factors were fitted to the $z$-score normalized relative abundance of each lineage. Here, monthly factors represent the proportional difference in frequency associated with that time-frame. The lineages were ordered by peak month. Key lineages are labeled. **C** Principal component analysis (PCA) of the $z$-score normalized relative abundance of the top 50 families within each sample ($n = 267$). Each sample is colored by collection month to illustrate the importance of seasonality for community composition. Data underlying the figure are provided in FigShare (https://doi.org/10.6084/m9.figshare.26082091).

## Microbial diversity

Overall, the microbial community was dominated by a few lineages, and the top 50 families constituted ~80% of all sequences (Fig. 2A). Taxonomic affiliation of each assembled contig and associated sequence reads were based on a consensus phylogenetic assignment using a majority rule, whereby the lineage at the lowest taxonomic rank to which at least 50% of open reading frames was assigned. The most common putative heterotrophic lineages included *Pelagibacteraceae*, *Rhodobacteraceae*, and *Flavobacteraceae*. The most common photosynthetic families were within cyanobacteria covering *Synechococcus* and *Prochlorococcus* as well as small eukaryotes like *Mamiellaceae* (incl. *Micromonas*) and the related *Bathycoccaceae* (incl. *Bathycoccus* and *Ostreococcus*). The metagenomic sequence proportions of *Prochlorococcus* and *Synechococcus* were strongly correlated with absolute cell count measured by flow cytometry at MiCRO ($R_{Pro} = 0.78$, $p < 1.7E-15$ and $R_{Syn} = 0.72$, $p < 2.8E-12$) (Supplementary Fig. 2). Further, when cell counts for these taxa were zero, gene proportions were also zero or near-zero, suggesting similar detection limits. Thus, there was a clear correspondence between the relative and absolute distribution of taxa.

## Seasonal succession

Common taxa showed clear seasonal succession (Fig. 2B). Seasonal (and interannual) anomaly was jointly calculated from fitting a linear model with monthly and yearly factors to the normalized proportion of each lineage. In the colder nutrient replete months (Dec to Feb), taxa including *Cytophagaceae*, *Alteromonadaceae*, *Oceanospirillacea*, and *Rhodobiaceae* as well as the photosynthetic *Bathycoccaceae* and ammonia oxidizers *Nitrosopumilaceae* all peaked in proportion (see "Data Availability Statement"). In the period with large increases in biomass and falling nutrients (March to May), *Flavobacteraceae*, *Pseudomonadaceae* and *Rhodobacteraceae* showed elevated proportions. During the summer, several non-Proteobacteria families including *Synechococcaceae* peaked. In the fall, *Pelagibacteraceae* proportion was high together with the photosynthetic *Prochlorococcaceae* and *Mamiellaceae*. The seasonal cycle in these taxa made a strong community imprint, and community composition clustered by collection month (Fig. 2C). In general, taxa that are more common under cold, nutrient replete conditions have large genomes and vice-versa for taxa more common during warm, nutrient deplete conditions. In support, we observed a clear oscillation in metagenomically-estimated average genome size. Here, we saw large genomes during the spring bloom period and the smallest at the end of the warm, oligotrophic period, when *Pelagibacteracea* and *Prochlorococcaceae* were dominating (Supplementary Fig. 3). Hence, linked to the seasonal taxonomic succession was an ecological succession between cells with larger versus smaller genomes.

Functional gene content demonstrated cycles with near clock-like precision and maximal separation between opposite seasons (Fig. 3). This was consistently seen across different functional classification schemes. We used a multi-table co-inertia analysis (MCOA) to quantify the overall covariance (i.e., shared principal components) across the functional classification schemes (COG, KEGG, Pfam and TIGRfam) plus taxonomic variation (family level). The MCOA analysis showed that the two primary principal components can explain approximately half of the shared functional and taxonomic variance (Fig. 3C). Furthermore, the top principal components showed clear seasonal oscillation (Supplementary Fig. 4). MCOA_PC1 (35% variance) peaked in the fall and thus was slightly offset from the temperature oscillation but more aligned with nutrient profiles. Hence, the maximum separation in community composition happened between spring and fall communities. MCOA_PC2 (16%) peaked in the summer and thus followed temperature. Visually, community functional potential clearly separated based on sample month (Fig. 3B, C). A comparison across all functional gene profiles reinforced these findings (Fig. 3D and Supplementary Fig. 5). Again, the temporal variation of each functional gene showed strong seasonality. Here, the first principal component (32% variance) again separated spring from late fall and the second principal component (11%) separated early summer from winter. Thus, the compositional succession of key taxa coincided with repeated oscillations in microbiome functional genes and overall potential.

## Interannual succession

Both taxonomic composition and functional potential showed an interannual succession tied to the ENSO climate cycle. The combination of seasonal and interannual shifts captured on average 62% of the total compositional variation (estimated using PERMANOVA, Supplementary Fig. 6). Interannual shifts in temperature correlated with the ENSO cycle (Fig. 4A). During La Niña conditions (negative ONI index), temperature was anomalously low at our site. This was seen in 2011–2013 and then again in 2020–2021. Strong El Niño conditions were present in 2014–2015 and to a lesser extent in 2018–2019. Most microbiome variation was seasonal, but interannual shifts explained an additional 14% of overall sequence proportions among all the genes. Interannual variability primarily partitioned into two clusters separating La Niña and El Niño conditions (Supplementary Fig. 7). Nearly half of this interannual variation (46%) was tied to a linear combination of temperature and nutrient availability (Fig. 4C). Thus, multi-year changes in temperature and nutrients made key impacts on this ecosystem. These interannual microbiome shifts mimicked the observed seasonality. As such, genes most frequent during cold, nutrient replete (La Niña) years also had a seasonal peak during the winter months

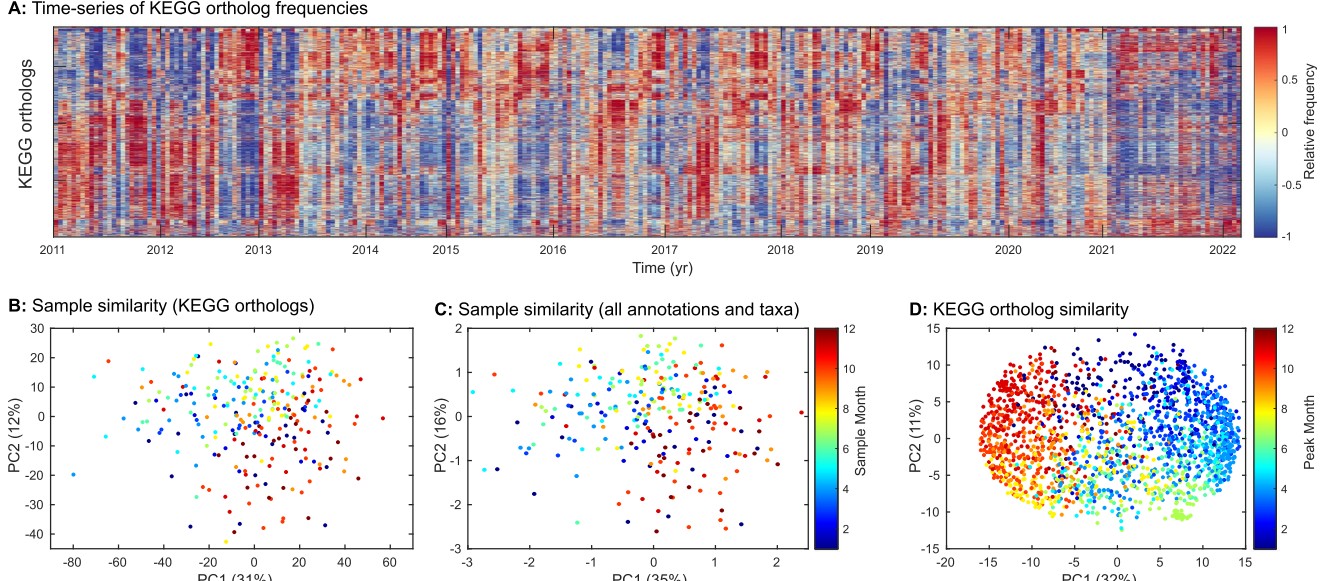

**Fig. 3 | Seasonal and long-term succession in microbial community functional potential. A** Time-series of functional genes annotated with the KEGG ortholog system. Time-series using other annotation systems (COG, PFam, and TIGRfam) show similar temporal patterns (Supplementary Fig. 5). The proportion of each gene was *Z*-score normalized across time. **B** Temporal dynamics of community functional potential. Sample similarity was estimated from *Z*-score normalized proportion of each KEGG ortholog using principle components analysis (PCA) to balance the weight of each gene. **C** Integrated community functional similarity using multi-table co-inertia analysis (MCOA). This integrated analysis used annotations from KEGG, COG, PFam, and TIGRfam as well as taxonomy. **D** Similarity in temporal dynamics of KEGG orthologs. The input data is the same as in this figure (**B**) but here we quantified the similarity in temporal dynamics of functional genes (rather than samples) using PCA. Each gene is colored by peak month again illustrating the importance of seasonal dynamics in structuring the variation in gene proportion. Data underlying the figure are provided in FigShare (https://doi.org/10.6084/m9.figshare.26082091).

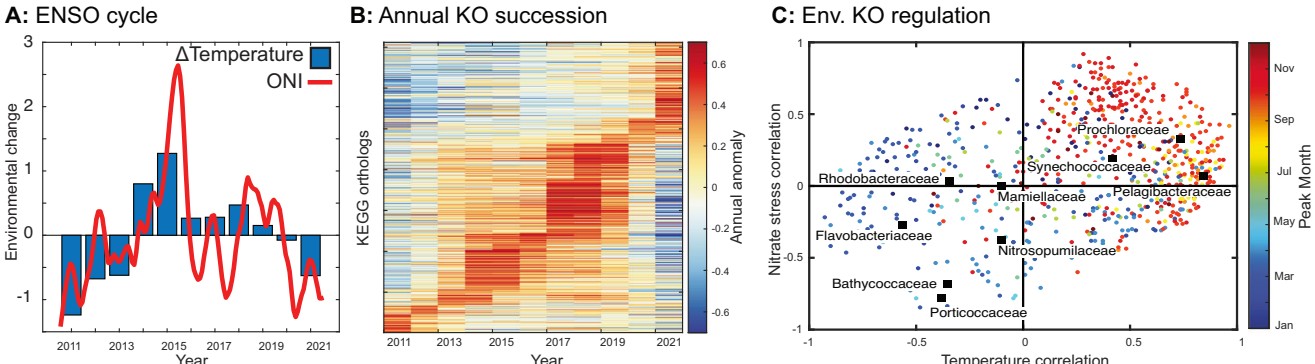

**Fig. 4 | Interannual changes in microbiome taxa and functions. A** Link between El Niño-Southern Oscillation (ENSO) phases (Ocean Nino Index, ONI, red line) and interannual temperature anomalies (blue bars) at MiCRO. **B** Interannual succession in microbiome functions. Interannual anomalies for each gene are sorted according to peak positive year. **C** Spearman correlation between interannual changes in temperature and nitrate stress (−1 * [nitrate]) vs. interannual anomalies for taxa or gene frequencies. Here, we show the interannual changes using KEGG annotations but similar patterns are seen using other annotation systems. Data underlying the figure are provided in FigShare (https://doi.org/10.6084/m9.figshare.26082091).

(Fig. 4C). Genes more frequent during warm, nutrient depleted (El Niño) years peaked during the late summer when similar conditions occurred. Functional gene shifts overlapped with biodiversity shifts. Cells with small, streamlined genomes were enriched during El Niño conditions (Supplementary Fig. 3). In contrast, the average genome size peaked during La Niña events including 2012 and 2020. Furthermore, *Prochlorococcacea*, *Synechococceae*, and *Pelagibacteraceae* peaked in the summer and fall but also in years with high temperature and low nutrients (Fig. 4C). In contrast, *Flavobacteraceae*, *Porticoccaceae*, and *Bathycoccaceae* peaked in the spring as well as during colder, nutrient replete years. Thus, the same environmental drivers control seasonal and interannual cycles of key lineages and functional potential.

## Functional succession

We next explored how the seasonal and interannual successions affected specific ecosystem and biogeochemical functions. A challenge is the presence of more than 10,000 unique functional genes each with individual annotations. To address this methodological obstacle, we trained a 'Natural Language Processing' mode using spaCy. We then weighted each individual annotation keyword by gene coverage by sample. We found that annotations including the term 'iron' were more common in the winter and spring. In contrast, annotations with the words 'nitrate', 'urea', and 'phosphatase' were more common in the summer and fall months. In past analyses of pelagic communities[11], we have detected a distinct biogeography of nutrient acquisition genes indicative of various forms of resource stress in phytoplankton. Seeing

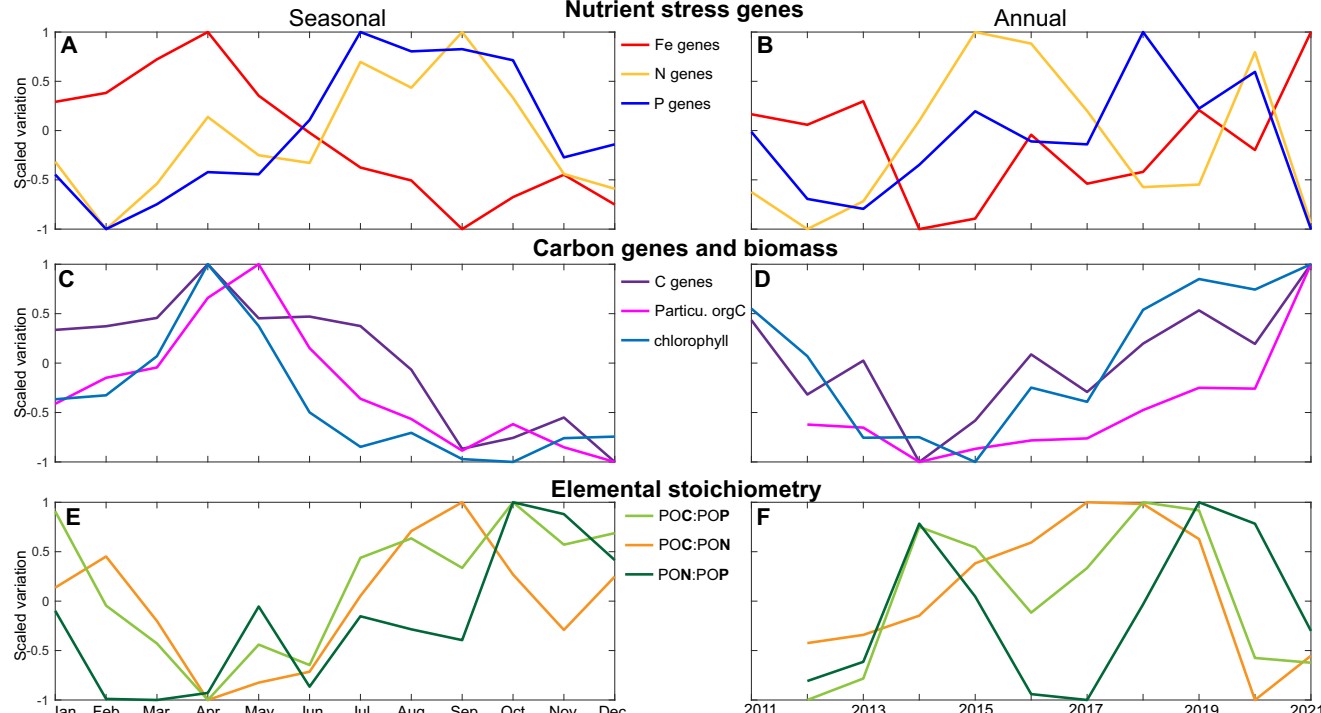

**Fig. 5 | Succession of biogeochemically important community aggregated traits at seasonal and interannual time-scale.** Scaled seasonal anomaly for (**A**) nutrient cycling genes, (**C**) carbon processing genes (CAZymes, i.e., carbohydrate degrading enzymes), particulate organic carbon (Particu. orgC/POC), and chlorophyll, and (**E**) elemental stoichiometry including C:P, C:N, and N:P. Scaled interannual anomaly for (**B**) nutrient cycling genes, (**D**) carbon processing genes, particulate organic carbon (POC), and chlorophyll, and (**F**) elemental stoichiometry including C:P, C:N, and N:P. Seasonal and interannual anomalies are scaled from −1 to 1. See Supplementary Data 2 for a list of genes linked to each biogeochemical category. See Supplementary Figs. 8 and 9 for the variation in genes for each biochemical function and a summary of phylogenetic affiliation. See Supplementary Fig. 10 for temporal variation in specific genes underlying each trait. Data underlying the figure are provided in FigShare (https://doi.org/10.6084/m9.figshare.26082091).

a parallel, albeit temporal, variation in annotation keywords led us to test if there were similar systematic shifts in biogeochemically important genes in this coastal community dominated by putative heterotrophic bacteria.

Following the exploratory keyword analysis, we next identified resource stress genes based on their annotations. These genes were responsible for inorganic nutrient transport, ability to access alternative resource forms including organically bound nutrients, and regulation (Supplementary Data 2). We then quantified the seasonal and interannual shifts at the levels of community aggregated traits (Fig. 5), specific biochemical functions (Supplementary Figs. 8 and 9), and individual genes (Supplementary Fig. 10). At the community aggregated trait level, the proportion of iron (Fe) stress genes were elevated in the winter and spring and peaked in April (Fig. 5A). The maximum proportion of this trait matched periods with high macronutrient supply in part due to upwelling (Fig. 1). This seasonal pattern is seen among multiple Fe acquisition genes covering both inorganic transport and the uptake of Fe bound to siderophores (Supplementary Data 2). Nitrogen (N) and phosphorus (P) stress genes increased in proportion later in the season (Fig. 5A). N stress genes included the biochemical switch to urea and nitrate use when ammonia is in low supply. P stress genes included inorganic phosphate transport and the use of organically-bound or reduced phosphorus. Both N and P stress community aggregated traits peaked between July and October, when macronutrient concentrations were at a minimum. A taxonomic analysis revealed that the Fe, N, and P community aggregated traits were mostly encoded by putative heterotrophs within Proteobacteria and Flavobacteria (Supplementary Figs. 8 and 9). Thus, we observed a clear seasonal succession in putative heterotrophic nutrient acquisition genes linked to the supply of macronutrients.

We also detected a parallel change in carbon availability and processing (Fig. 5C). Glycoside hydrolases and supporting functions

form an important trait for the degradation of large organic molecules[27–29]. This organic matter degradation trait constituted 3.6% of all annotated sequences and peaked in proportion in the spring. The observed seasonal pattern matched the period with high biomass represented by both chlorophyll and particulate organic carbon (POC). Furthermore, chlorophyll and POC declined in tandem with nutrient concentrations in the summer and reached a minimum in the fall. This was followed by a similar decline in the proportion of carbon degradation enzymes.

**Linking community and biogeochemical succession**
In this ecosystem, plankton constitute a large fraction of the overall particulate organic matter[16]. Thus, we hypothesized that the succession in key microbial biogeochemical functions would make an imprint on the organic matter carbon-to-nutrient ratios, which are commonly high during N or P limitation[30]. In support, we observed that both the carbon-to-nitrogen (C:N) and carbon-to-phosphorus (C:P) ratios were low in the winter and spring when POC was high, macronutrients were high, and N and P acquisition genes were low in sequence proportion (Fig. 5E). In contrast, we observed high C:N and C:P ratios in parallel to the higher frequencies of N and P acquisition genes in the summer and fall. Thus, we detected a clear parallel succession between key community traits and the carbon-to-nutrient ratios in organic matter.

Similar to seasonal shifts, we detected a link between interannual shifts in key community traits and nutrient and carbon biogeochemistry. The sequence proportion of Fe stress genes were elevated during the cold, nutrient replete 2011–13 La Niña years, dropped during the El Niño event in 2014 and 2015, and reached a maximum during the next La Niña in 2021 (Fig. 5B). N stress was anti-correlated to Fe stress ($r_{spearman} = -0.67$) and peaked during the El Niño event. P generally followed N stress but reached a maximum in 2018. Organic matter

degrading enzymes correlated positively with concentration of chlorophyll and POC with maxima during La Niña and a minimum during El Niño events (Fig. 5D). These patterns paralleled shifts in the C:N:P ratios (Fig. 5F). Here, all ratios were low during the La Niña periods and mostly elevated between 2014 and 2019. However, C:N peaked in 2018, whereas C:P and N:P peaked the next year. Hence, the biogeochemical dynamics at the interannual scale was similar to the seasonal scale with colder months matching colder years and vice-versa. This gave rise to a clear ENSO-driven cycle between La Niña years with high macronutrients, Fe stress, and organic carbon processing genes, but low N and P stress genes. The opposite pattern was seen during the El Niño event. Thus, the ENSO cycle led to clear microbiome shifts impacting the biogeochemical coupling between ocean carbon and nutrient cycles.

## Discussion

Our results clearly support a strong link between marine microbiome compositional and functional dynamics at both seasonal and interannual time-scales. One common model posits that while individual taxa are sensitive to environmental change, genetic redundancy leads to resilience in overall microbiome functions[10]. In contrast, common lineages including *Prochlorococcus*, *Synechococcus*, and *Pelagibacter* are all subject to extensive gene gain and losses resulting in connected global phylogenetic and functional biogeographies[31,32]. Furthermore, key lineages display extensive seasonal niche differentiation[4,19]. A time-series from the Mediterranean Sea[9] and now our MiCRO time-series observations suggest parallel taxonomic and functional microbial biodiversity cycles. Given that metagenomic analyses represent the relative proportion of taxa and not directly absolute changes[33], there are alternative explanations for the observed patterns at MiCRO. Our results may be explained by a potentially stable 'background' oligotrophic community with dynamic shifts in copiotrophic lineages 'pushing down' the proportion of other lineages and their associated functional profiles. This model mimics phytoplankton dynamics observed during a diatom bloom[34]. Alternatively, a strong correspondence between relative and absolute changes would signify metagenomic proportions that are indicative seasonal and interannual shifts of all community members. At MiCRO, we detect a clear correspondence between flow cytometry counts and metagenomic proportions (Supplementary Fig. 2). This correspondence may be driven by limited annual changes in total bacterial abundance, which typically varies by a magnitude of 2–3×[35]. Indeed, evidence suggests that mean prokaryotic biomass varies by just under 3× in the surface ocean across temperate and subtropical marine ecosystems[36–38]. Moreover, *Prochlorococcus* is often regarded as a model of oligotrophic adaptive strategies[39], and this lineage is undetectable with flow cytometry at MiCRO when the metagenomic proportion is low[18] (Supplementary Fig. 2). Combined, these observations suggest a substantial connection between absolute and relative changes at MiCRO and challenge the notion of functional redundancy in marine microbiomes. Similar to phytoplankton succession in many aquatic ecosystems[40–42], we find evidence of seasonal and interannual succession in microbiome lineages and functional potential.

The observed succession in biodiversity and functional potential is tied to ocean biogeochemical cycles. Spatially, Fe stress is connected to elevated vertical nutrient supply, whereas N and P stress are found in warm, highly stratified regions[43]. Even regions not commonly regarded as Fe-limited like the Iceland Basin can also experience seasonal Fe stress following periods of high nutrient supply[44]. In contrast, macronutrient stress is generally controlled by temperature and stratification and thus peaks in the summer or early fall[45]. Our observed seasonal patterns in nutrient stress are consistent with these spatial and temporal dynamics in other systems. ENSO-driven changes to the Equatorial Pacific Ocean revealed a strong positive correlation between temperature and Fe stress[46] and thus the opposite of our

observations. However, reduced Fe stress during El Niño events is consistent with stratification and lower macronutrient flux[47]. These observations of nutrient limitation in other regions are derived from phytoplankton dynamics. However, cycles in nutrient stress genes at MiCRO were primarily occurring in putative heterotrophic bacteria. We know less about nutrient limitation in heterotrophic organisms. In another microbiome time-series from the Mediterranean Sea, a spring peak in bacterial Fe stress gene frequencies was also observed[9,48]. Furthermore, there are reports of Fe stress in heterotrophic organisms in HNLC regions[49] and N and P stress in warm, stratified regions[50]. In the California Current, Fe additions only stimulated bacterial growth when added together with carbon substrates[51]. This suggests some level of regional bacterial Fe stress, but perhaps not direct Liebig-style limitation. Thus, the microbiome time-series showed functional shifts among mainly heterotrophic microorganisms and matched biogeochemical theory.

Using biomarkers to identify ecosystem state is based on the process of bacteria adapting to specific environmental conditions via gene gain and loss[52,53]. Evolution can occur from the selection and rise of less common genotypes drawn from a large 'rare biosphere' among microbial communities[54]. Alternatively, cells may actively gain and lose genes via lateral transfer[55]. Independently of the exact mechanism, we can detect shifts in key gene stoichiometries conferring adaptive advantages to local environmental conditions[11,56]. Here, the chosen biomarkers represent 'entry points' for resource acquisition like membrane transporters or conversion of alternative resources into inorganic versions (e.g., organic P to phosphate). We observe temporal increases in some transporters (e.g., for phosphate) that presumably are essential when nutrient resources are scarce. This may be due to key marine lineages having duplicate copies of nutrient transporters when proliferating in low resource environments[57,58]. As reflected in their expression regulation, such nutrient acquisition mechanisms are dispensable if more attractive alternatives are present (e.g., ammonia instead of nitrate)[59]. In contrast, glycoside hydrolases are normally upregulated during resource presence (e.g., cellobiose[60]). The distribution of organic matter degradation biomarkers shows a parallel pattern to such a regulatory control. Nutrient stress biomarkers reflect a resource depletion, whereas the organic matter degradation biomarkers reflect a resource presence. Hence, the presence of high POC may select for organisms with these genes. Our biomarker concept also requires rapid selection and is sensitive to dispersal of genotypes adapted to adjacent locations[61]. For example, the MiCRO site is located in a near-shore environment, where sediment suspension should lead to a high iron flux. It is unexpected to detect Fe stress under such near-shore conditions, but intermittent iron stress has been seen further off-shore in the California Current ecosystem[62,63]. The local ocean circulation is very dynamic with currents exceeding 0.1 m s$^{-1}$[64,65], so we are likely observing cells adapted to conditions in the wider region. Other approaches like metatransciptomics would enable a finer-scale characterization and thus supplement the presented genomic biomarker approach.

Our observations suggest a strong climate sensitivity of marine microbiome biodiversity and functions. Past studies have observed seasonal oscillations in rRNA diversity in many ocean environments[25,35,66]. Both the seasonal and ENSO climate cycles drive oscillations between cold, nutrient rich and warm, nutrient deplete conditions[67]. Such environmental shifts broadly resemble conditions associated with anthropogenically-driven climate change. Specifically, observed tradeoffs between high biomass, Fe-limited conditions and low biomass, N- and P-limited conditions (Fig. 5) are consistent with modeled projections of climate-driven shifts between Fe- and N-limitation and associated impacts on eastern equatorial Pacific ecosystems[68]. Thus, the consistent microbial response observed across both seasonal and interannual time-scales points towards large climate-driven changes in biodiversity, ecosystem functions, and

biogeochemistry. For the SCB, this means an ecosystem increasingly dominated by cells with small genomes including *Prochlorococcus* and *Pelagibacter*, declining organic carbon degradation and POC concentrations, less iron but more macronutrient stress, and cells with elevated carbon-to-nutrient ratios. Such changes will likely have wide impacts on the broader ecosystem.

## Methods

### Sample collection
Sample collection occurred at the MiCRO time series at Newport Pier in Newport Beach, California, USA (33.608°N and 117.928°W) between September 2009 and 2021 at daily to monthly (weekly, on average) temporal resolution. Sampling protocols have been previously described[16,18,21]. Four autoclaved bottles were rinsed with nearshore surface water before collection and immediate transport to the lab. A summary of all metagenomic samples is listed in Supplementary Data 1.

A total of 267 DNA samples (replicated) were collected through filtration of 1 L of seawater through a 2.7 μm GF/D and a 0.22 μm polyethersulfone Sterivex filter (Millipore, Darmstadt, Germany) using sterilized tubing and a Masterflex peristaltic pump (Cole-Parmer, Vernon Hills, IL). DNA was preserved with 1620 μl of lysis buffer (23.4 mg/ml NaCl, 257 mg/ml Sucrose, 50 mM Tris-HCl, 20 mM EDTA) and stored at −20 °C before extraction.

Particulate organic matter (POM) was collected using two autoclaved bottles. Particulate organic carbon/particulate organic nitrogen (POC/PON) and particulate organic phosphorus (POP) were each collected by filtering 300 ml of seawater through pre-combusted (500 °C, 5 h) 25 mm GF/F filters (Whatman, MA). POC and PON were collected on the same filter. POC/PON and POP both had six replicates collected, a triplicate for each bottle. After filtration, samples were placed on petri-dishes and stored in a −20 °C freezer.

Nutrients were collected from the filtrate of the particulate organic matter (POM) sampling. Sample water was filtered through a 25 mm GF/F with a nominal pore size of 0.7 μm (used for POM) and was then re-filtered through a 0.2 μm syringe filter into a prewashed 50 ml tube. Filtrate was then stored in a −20 °C freezer before the determination of nitrate concentration and phosphate as soluble reactive phosphorus (SRP) concentration.

### Shore-station measurements
Temperature and chlorophyll were recorded via the Southern California Coastal Observing Systems (SCCOOS) automated shore station (Newport Pier) located at the sampling site. Sensors include a Seabird SBE 16plus SeaCAT Conductivity, Temperature, and Pressure recorder and a WetLabs WetSTAR sensor for chlorophyll. Measurements were taken at 4 min resolution, but we transformed the data into daily averages to align with other environmental measurements. The ENSO was measured using the Ocean Nino Index v5 (ONI) from the National Weather Service Climate Prediction Center.

### Nitrate measurements
From 2011 to 2018: Nitrate samples were thawed in a refrigerator overnight. 50 ml standards were created from an artificial sea water and potassium nitrate solution (10 μM). Standards ranged from 0 μM to 10 μM concentration of potassium nitrate solution across 10 vials. 500 μl of ammonium chloride solution (4.7 M) was added to each 50 ml sample and standard and mixed. A sample was poured into a column of copperized cadmium fillings[21,69] and 15 ml was used to flush the system. 25 ml was then collected from the column and 500 μl of sulfanilamide solution was added to the sample, mixed, and allowed to react for 6 min. 500 μl of N-(1-Naphthyl)-ethylenediamine dihydrochloride solution was then added to the tube, mixed, and allowed to react for 20 min. The sample was read on a spectrophotometer set at a wavelength of 543 nm[70]. From 2018 to 2019: Samples were processed using a QuickChem FIA 8500 autoanalyzer (Lachat Instruments, Loveland,

Colorado, USA), with a detection limit of 0.014 μmol. Beginning in 2019, samples were processed with spongy cadmium. Spongy cadmium was created with a cadmium sulfate solution (10 g CdSO4 in 50 ml DI water). Zinc sticks are added to the solution and allowed to sit for 8 h. After sitting, the zinc sticks were washed with 6 N HCl, along with several drops of 6 N HCl added to the CdSO4 solution and drained. Precipitated cadmium was covered with 6 N HCl and stirred to break up the cadmium. The cadmium was drained and rinsed 10 times with DI water. The cadmium in this state was used for this protocol. 5 ml of sample was placed in borosilicate glass tubes (acid-washed) with 1 ml of NH4Cl solution (0.7 M) added to each. 0.2 g of spongy cadmium was added to each and then capped off, laying horizontally on a mechanical shaker table (100 excursions/ minute) for 90 min. 5 ml of sample was pipetted out of the tube and placed in a disposable culture tube. 250 μl of color reagent (mixture 5 g sulfanilamide, 0.5 g N-(1-Naphthyl)-ethylenediamine dihydrochloride mixed in 50 ml phosphoric acid (8.51 M), diluted in 500 ml of nanopure water) was added, vortexed, and allowed to react for 10 min. The sample was then read using a spectrophotometer set at a wavelength of 540 nm. Samples were compared to a set of six standard solutions containing artificial sea water and diluted 100 μM KNO$_3$ (standard KNO$_3$ concentration: 0 μM to 60 μM). Standards were treated the same as samples[70].

### Phosphate measurements
SRP concentrations were determined using the magnesium induced co-precipitation (MAGIC) protocol[71,72]. SRP samples to be processed were moved to the refrigerator overnight to thaw. Once thawed, 0.4 ml of 3 M NaOH was added to each tube and shaken vigorously. After 5 min the samples were placed into a centrifuge set at 1500 g for 20 min. Supernatant (P-free seawater) was poured into a separate glass (needed for standards) from the sample tubes and allowed to dry for an hour. 6 ml of 0.25 M HCl was added to all tubes and shaken to dissolve each pellet. 0.66 ml of Arsenate (2:2:1 parts sodium metabisulfite (0.74 M), sodium thiosulfate (88.5 mM), sulfuric acid (3.5 N)) was added to each tube and allowed to react for 15 min in the dark. 0.7 ml of mixed reagent (2:5:1:2 parts ammonium molybdate tetrahydrate (24.3 mM), sulfuric acid (5 N), potassium antimonyl tartrate (4.1 mM), and ascorbic acid (0.3 M)) was added to the tubes and allowed to react for 30 min in the dark. Samples were then read on a spectrophotometer set to a wavelength of 885 nm, using a 0.125 M HCl solution as black and rinsing agent. Ten standards were made using the P-free seawater collected from the samples and mixed with 1 mM KH$_2$PO$_4$ creating different dilutions (2 nM to 1 μM KH$_2$PO$_4$). Standards were treated the same as the samples once made. From 2018 to 2019 samples were sent out to be processed on an auto-analyzer. These samples were also run in parallel with samples using the MAGIC protocol for comparison.

### Particulate organic matter
POC/N samples were processed according to Sharp (1974)[73]. Filters for POC/N were dried at 55 °C for 24 h and then wrapped with tin discs (CE Elantech). The wrapped filters were combusted in a FlashEA 1112 (Thermo-Scientific) using the NC Soils setup. The oxidation reactor was set to 900 °C, the reduction reactor was set to 680 °C, and the oven was set to 50 °C. Oxygen gas (UN 1072, Airgas) was injected at a flow rate of 250 ml/min, allowing sample combustion to occur at 1800 °C. Helium gas (UN 1046, Airgas) was used as the carrier gas with the carrier flow rate set to 130 ml/min and the reference flow rate set to 100 ml/min. The compounds serving as standards were acetanilide (71.1% C, 10.4% N) and atropine (70.6% C, 4.8% N). The minimum detection limits for this analysis were 2.4 μg C and 3.0 μg N.

POP was analyzed using a modified ash-hydrolysis protocol[72]. Samples were removed from the freezer and placed in combusted scintillation vials; along with a set of K$_2$HPO$_4$ (1.0 mM-P) to be used as standards. 2 ml of MgSO$_4$ (0.017 M) was added to each vial, covered in

tin foil, and placed in an 80 °C oven overnight. The vials were moved to a 500 °C combustion oven for 2 h. Once cooled 5 ml of HCl (0.2 M) was added to each vial and put back into the 80 °C oven for 30 min. The solution in the vials was transferred to a 15 ml centrifuge tube. Each vial was rinsed with 5 ml of DI water, which was transferred to the respective centrifuge tube. 1 ml of mixed reagent [2:5:1:2 parts ammonium molybdate tetrahydrate (24.3 mM), sulfuric acid (5 N), potassium antimonyl tartrate (4.1 mM), and ascorbic acid (0.3 M)] was added to each centrifuge tube in 30 s intervals and moved to the dark. Tubes were centrifuged for 2 min to concentrate any glass fibers or debris to the bottom of the tube. After 30 min samples were then analyzed using a spectrophotometer set to a wavelength of 885 nm, with a HCl (0.1 M) solution as a blank and rinsing agent between samples.

### Environmental data analysis

Environmental data was collected at higher sampling frequency compared to DNA. We included all data points to reduce uncertainty for monthly and annual estimates of variation. We fitted a linear model to all data points using 12 monthly and 12 annual (2011–2021) categorical factors. This was done in Matlab with 'anovan'. Moving averages were calculated with robust loess to lower the weight of outliers (using Matlab 'rloess').

### DNA extraction

To extract DNA, we use an adapted protocol[74]. Sterivex filters were first incubated at 37 °C for 30 min with lysozyme (50 mg/ml final concentration). Next, Proteinase K (1 mg/ml) and 10% SDS buffer were added and incubated at 55 °C overnight. A solution of ice-cold isopropanol (100%) and sodium acetate (245 mg/ml, pH 5.2) was used to precipitate the released DNA, which was then pelleted via centrifuge and resuspended in TE buffer (10 mM Tris-HCl, 1 mM EDTA) in a 37 °C water bath for 30 min. DNA was purified using a genomic DNA Clean and Concentrator kit (Zymo Research Corp., Irvine, CA). Finally, extracted DNA was stored at −80 °C.

### DNA sequencing

DNA concentration was assessed using a Qubit dsDNA HS assay kit and a Qubit fluorometer (ThermoFisher, Waltham, MA). Next, 30–60 ng of genomic DNA per sample was visually examined via agarose gel electrophoresis to check for degraded DNA. Finally, a Nanodrop ND-1000 (ThermoFisher, Waltham, MA) was used to assess sample purity and verify that A260/A280 values were between 1.6 and 2.0 and A260/A230 values were between 2.0 and 2.2. Frozen DNA was transported in 25–500 µl of 1×TE DNA suspension buffer per sample to the DOE Joint Genome Institute (JGI).

A total of 236 samples collected between 2011 and 2020 underwent successful short-read metagenomic sequencing at JGI. After QC, total of 120 samples required additional size selection, thus, an input of 10.0 ng of genomic DNA was sheared around 300 bp using the LE220-Plus Focused-ultrasonicator (Covaris) and size selected with a double SPRI method using Mag-Bind Total Pure NGS beads (Omega Bio-tek). Using the KAPA-HyperPrep kit's (Roche) one-tube chemistry of end-repair, A-tailing, and ligation with NEXTFLEX UDI Barcodes (PerkinElmer), the sample was enriched using 7 cycles of PCR. For all samples, the prepared libraries were quantified using KAPA Biosystems' next-generation sequencing library qPCR kit and run on a Roche LightCycler 480 real-time PCR instrument. Sequencing of the flow cell was performed on the Illumina NovaSeq sequencer using NovaSeq XP V1.5 reagent kits, S4 flowcell, following a 2 × 151 indexed run recipe. A total of 3.21 Tbp of short-read metagenomic data was produced at an average of 13.6 Gbp/sample, exceeding target sequencing depth.

An additional 31 metagenomic libraries were sequenced from samples collected in 2021–2022 (total 0.259 Tbp, average 8.36 Gbp/sample), this data is available through the National Center for Biotechnology Information Sequence Read Archive (BioProject ID PRJNA624320). For sequence library preparation please see 'Supplementary Information'.

### Assembly and annotation

Short-read metagenomes were processed following the Joint Genome Institute (JGI) metagenomics workflow[75]. JGI's pipeline can be implemented using the National Microbiome Data Collaborative's open-source online platform Empowering the Development of Genomics Expertise[76]. For specific details on metagenomic assembly and annotation, please see 'Supplementary Information'.

### Average genome size

The average genome size was calculated according to the standard JGI IMG pipeline[75]. Briefly, the estimate is calculated based on counts of 139 single copy marker genes identified in ref. [77] from the assembled metagenome. Each marker gene is represented by a unique PFAM domain annotated within IMG and is calculated as: $N_{Bases\_assembly}/N_{Genomes}$. There is an additional check on the Coefficient of Variation (= (Std Dev)/(Mean)). If the coefficient of variation is greater than or equal to 50%, then an estimated average genome size is not considered reliable and is not reported.

### Community analysis

The microbiome time-series analysis was based on many of the recommendations from Coenen et al. [78]. The unit of biodiversity (akin to an OTU) is either taxonomic (at 'Family' level) or unique protein families using primarily KEGG Orthologs (KO) but also COG, Pfam, or TIGRFAM classifications. Occurrence was normalized to summed total coverage for each sample. We excluded biodiversity units with a relative abundance below 5E-5 resulting in a low occurrence of absent units (i.e., zeros). Only having a few instances of zeros for the relative proportions allowed us to use linear models (e.g., PCA) rather than applying non-linear tools (like Bray-Curtis similarity and PCoA) when comparing samples or units of biodiversity. All community analyses were done using Matlab unless otherwise noted.

We normalized each biodiversity unit to a mean of zero and a standard deviation of one (i.e., z-score) when analyzing the temporal variation of individual taxa or genes. Anomaly was next quantified by fitting linear models with monthly (12 levels representing seasonal anomaly) and yearly (12 levels representing interannual anomaly) categorical factors and no interactions using Matlab 'anovan'. A low frequency band-pass filter using 'lowess' (Matlab 'smooth' function) was applied when assessing temporal changes. There was a strong autocorrelation with temporally adjacent samples sharing 92.5% taxonomic and 98.7% functional similarity.

### Compositional variance analysis

We used PERMANOVA[79,80] to quantify whole community composition variance associated with seasonal and interannual changes. We used sampling month and year as well as their interactions as factors. Community composition was estimated using taxonomic (Family level) or functional (COG, KO, Pfam, and TIGRfam) changes. We estimated the Euclidean distance of the z-score normalized sequence proportion of each unit of biodiversity for the compositional matrix. We used 'adonis' from the 'vegan' package (v2.6-4) in $R$[81,82] for this analysis.

### Multi-table co-inertia analysis

A joint principal component analysis of both taxonomic (Family level) and functional changes (COG, KO, PFam, and TIGRFam) was performed as previously described[83]. In brief, we used Multi-table Co-Intertia Analysis from the 'ade4' package (v1.7-22) in $R$[84,85]. The first two principal components are presented from this analysis. Similar to

regular principal component analyses, the MCOA analysis relied on linear distances between samples and taxa or functional gene groups.

## Annotation keyword model

A 'Natural Language Processing Model' was trained on all gene annotations. This was done using the 'spaCy' Python package[86]. The model treats all verbs and nouns as tokens and classify them into bins. Thus, each annotation keyword can be considered akin to an OTU. Subsequently, we calculated the time-series of all keywords weighted by the relative abundance of the associated gene using the same analysis pipeline as for the community data (i.e., fitting linear models to estimate anomaly and temporal changes). Next, we found the seasonal oscillation and annual change of each keyword from the Sum of Squares following the linear model fit. Finally, we identified all KOs and Pfams with keywords showing high seasonal and annual variation.

## Biogeochemical genes

Following the keyword analysis, we used the KEGG database to identify all genes responsible for Fe, N, and P acquisition (Supplementary Data 2). For Fe, these genes encoded transporters for uptake of inorganic forms as well as iron bound to siderophores. For N, the functions covered nitrogen fixation, urea uptake and hydrolysis, and nitrate and nitrite uptake [both using NADH (most bacteria) and Ferredoxin (mainly Cyanobacteria)]. For P, genes encoded inorganic phosphate transporters, alkaline phosphatase (cleaving P off phospho-esters), and phosphonate transporters and degradation. We identified the associated KEGG Ortholog IDs and retrieved their individual (Supplementary Figs 8–10) and summed coverage for each sample (Fig. 5). We used Pfam to identify genes involved in degrading organic matter degradation identified previously[28]. These genes cover carbohydrate polymer degradation, binding and uptake. We retrieve the individual and summed coverage for samples using the associated Pfam IDs. To calculate the community aggregated trait proportion, we summed the coverage of all Fe, N, P, and carbon genes, respectively. We next calculated the monthly and yearly anomaly of the summed traits (Fig. 5), biochemical function (Supplementary Figs. 8 and 9) and individual genes (Supplementary Fig. 10). This was done using a linear model with 12 monthly and 12 yearly levels using Matlab 'anovan'. The individual gene families were grouped according to their biochemical function.

We evaluated the quality of functional assignments of these biogeochemical genes by comparing annotations using KOs, COGs, ECs, SMART, and Pfam. First, we observed a high degree of consistency in functional description across annotation systems. We observed gene fragments within a single KO had a single annotation in 71–100% of cases with other systems. Second, only ~10% of the analyzed KOs were annotated with different functions using these other systems. This likely reflects the uncertainty in annotations with any specific system. Thus, other annotation pipelines resulted in supporting—and uniform—annotations.

The phylogenetic origin of each gene family (i.e., KO or Pfam) was determined as the consensus phylogeny for the associated contig (see 'Supplementary Information'). These were weighted by the contig coverage, and the summed phylum coverage for each gene family was calculated using Matlab 'GroupSummary'. These values were normalized to total (across phylum) coverage by gene family.

## Reporting summary

Further information on research design is available in the Nature Portfolio Reporting Summary linked to this article.

## Data availability

Environmental data can be accessed via the MiCRO BCO-DMO data page (https://doi.org/10.26008/1912/bco-dmo.564351.2)[87]. Metagenomic sequencing data is available through the National Center for Biotechnology Information Sequence Read Archive (BioProject IDs: PRJNA1220992, PRJNA624320). For SRA Accession IDs (i.e., SRP and SRR) please see Supplementary Data 1. Metagenomic data and analysis products, including annotations, are available through the Joint Genome Institute Genome Portal (https://doi.org/10.46936/10.25585/60001365)[88]. Data underlying the figures, including MiCRO metagenome gene and taxon frequencies and analysis, are provided in FigShare (https://doi.org/10.6084/m9.figshare.26082091)[89].

## Code availability

All code applied to this analysis used standardized packages and platforms (Python, R, and Matlab). For inquiries regarding the code described in this manuscript, please contact the corresponding author.

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

## Acknowledgements

This work was partially funded by the U.S. Department of Energy Joint Genome Institute Community Science Program 2021. The work (https://doi.org/10.46936/10.25585/60001365 to A.A.L. and A.C.M.) conducted by the U.S. Department of Energy Joint Genome Institute (https://ror.org/04xm1d337), a DOE Office of Science User Facility, is supported by the Office of Science of the U.S. Department of Energy operated under Contract No. DE-AC02-05CH11231. We also acknowledge support from the National Science Foundation (OCE-2135035 to A.C.M.), National Oceanic and Atmospheric Administration (NA19NES4320002 to A.C.M.), the National Aeronautics and Space Administration (80NSSC21K1654 to A.C.M.), and the National Institutes of Health (T32AI141346 to M.L.B.).

## Author contributions

A.C.M. and A.A.L. conceived the project and contributed to securing funding, analyzing results, and writing the manuscript. A.A.L., M.L.B., A.J.F., A.R.M., S.D.G., L.E.L., and S.A.S. contributed to sample collection, processing, and/or the production and submission of data or metadata. E.A.E.-F. oversaw sequencing and production of data through bioinformatics pipelines at JGI. M.L.B. oversaw additional sequencing and bioinformatic analysis. All authors contributed to the discussion of results, manuscript preparation, and revision.

## Competing interests

The authors declare no competing interests.
