## [Transparent Peer Review file · Nature Communications]

Climate-driven succession in marine microbiome biodiversity and biogeochemical function

Corresponding Author: Dr Adam Martiny

Version 0:

Reviewer comments:

Reviewer #1

(Remarks to the Author)

This study uses contemporary seasonal and interannual time series data from the southern California Current (metagenomic, biogeochemical, biological, physical) to provide insights into the effects of a changing ocean on the base of the marine food-web. The authors revealed seasonal cycling of marine microbiome function linked to key microbial functional traits and environmental variability over more than a decade. This allowed identification of elevated nutrient stress and decreased organic carbon cycling through the functioning of the microbiome under El Nino conditions (which resemble predicted future conditions), reflecting a shift towards oligotrophic taxa, traits, and functions. The results are noteworthy, as previous studies have lacked the long-term temporal resolution provided here or have been limited by only focussing on microbial taxonomic diversity, rather than function. However, I feel the use of broad classifications of functional genes (iron, nitrogen, and phosphorous stress genes, and carbon genes), potentially masks some of the complexity and intricacies of the ocean microbiome and the biotic and abiotic interactions that determine ecosystem function.

The authors aimed to “establish a genome-enabled understanding of the climate driven feedbacks...” (lines 70-72), however the metagenomic analyses are gene centric. Functional genes were linked to specific taxonomic ranks, but not genomes of individuals or populations of microorganisms. The large number of samples and the range of coverages of individual taxa provided by the highly seasonal nature of the site, should assist the authors in recovering genomes from these samples?

While the nuances of iron, nitrogen, and phosphorous cycling genes are demonstrated in the Extended Data Figure, they are discussed in rather a coarse manner in the main manuscript. For example, on lines 226-233, seasonal patterns in “stress genes” and their links to macronutrient supply are detailed, but no further information is provided as to the nature of the stress or the form of iron, nitrogen, or phosphorous that is inferred to be limiting from the functional genes. This idea is planted on line 211 in regards to previous work on phytoplankton, but not thoroughly explored in my opinion. It is also unclear in the text whether the authors have examined whole pathways associated with “nitrogen stress” for example, and how have they have treated incomplete pathways or poorly annotated pathways.

A similar comment for the carbon cycling section of the manuscript. I felt it was lacking in detailed information about the types of carbon and potential sources/interactions and seasonality that could be inferred from the identity and phylogeny of specific genes associated with carbon availability and processing.

Some of the methodological details are lacking which may limit complete reproducibility. The authors mention some analyses performed in Matlab but do not provide information about the commands used (line 438). They also mention the use of custom hidden Markov models that were implemented for the functional annotations but not what they were targeting (line 493). It is unclear what cut-offs were used to assign functional gene annotations. The authors used a keyword model to determine seasonal patterns in the frequency of specific functional genes based on their annotations as an exploratory tool – this of course can be flawed if annotations are incorrect/incomplete etc. Were further steps taken to confirm the presence of functional motifs in gene sequences associated with key annotations?

The Extended Data Figures provide useful information in support of the authors conclusions. In Extended Data Figure 1, it would be helpful to highlight the ENSO cycle on these plots. In Extended Data Figure 6, some indication of what the x and y axis represent on the heatmaps (right hand side of the figure) is needed. In Extended Data Figure 7 and 8, additional descriptions of these figures are missing. It would be useful to highlight that the genes included within each pathway are

detailed in the Supplementary Table.

Overall, the research is of significance to the field of marine science, and will be of interest to climate scientists, microbiologists, and biological oceanographers, as well as more broadly to researchers with an interest in time-series analyses, ecology and biogeochemistry. The manuscript is well-written and interesting, and the authors have succinctly reported on what is clearly a large and complicated dataset.

Reviewer #2

(Remarks to the Author)

This manuscript by Larkin et al. presents a comprehensive and innovative analysis of an important 11-year metagenomic dataset from the Southern California Bight. The study offers valuable insights into the seasonal and interannual dynamics of marine microbial communities and their functional potential, particularly in relation to ENSO cycles. This work has the potential to make a substantial contribution to our understanding of climate-driven changes in marine microbial ecosystems. While the research is well-conducted and the manuscript is well-written, there are some concerns regarding data interpretation and presentation to be addressed.

Major Concerns

A primary concern with this study is the interpretation of relative abundance data and the use of anomalies as a metric. The authors describe oligotrophic functions as increasing or decreasing in frequency in summer and fall, but this may not accurately reflect absolute abundances. It's possible that these functions maintain a consistent absolute abundance year-round, only appearing "depleted" when copiotrophs increase.

This study, like many omics analyses, relies on relative abundance data, which can lead to misleading interpretations. While the authors use terms like increased or depleted frequency in some instances, they often imply actual increases or decreases in abundance, which cannot be inferred from relative data alone.

An alternative interpretation could be that oligotrophic taxa maintain stable abundances throughout the year, while copiotrophic taxa increase during spring's heightened primary production. This scenario would show the same relative abundance patterns but with very different ecological implications.

The use of the term "succession" is particularly problematic, as it implies species displacement over time as niches evolve. It's unclear if this accurately describes the dynamics in the Southern California Current. Are copiotrophs truly displacing oligotrophs, or are they simply increasing in abundance alongside a stable oligotrophic population?

To address these limitations, the authors should discuss the constraints of relative abundance analysis and using anomaly as a metric. Additionally, incorporating total cell abundance data, if available, could provide crucial context. This would help distinguish between actual changes in community composition and increases in copiotrophic taxa overlaying a consistent oligotrophic population. If such data were collected, normalizing to total abundances could reveal which taxa and functions remain constant throughout the year and which exhibit true dynamics.

Here are several lines that exemplify this:

Ln 69 - "Thus, we predict that El Niño-driven warming will result in increases of oligotrophic bacterioplankton taxa with a concurrent increase in metabolic strategies associated with oligotrophic communities."

Ln 122 "These common taxa showed clear seasonal succession (Fig. 2B)."

LN 129 "In the fall Pelagibacteraceae was common..." This implies that Pelagibacteraceae was not common in the other seasons, which seems highly unlikely. How often in the dataset are the Pelagibacteracea in the top 3 most abundant families, if not the very top. This is often the case in most marine systems.

LN 178 This statement wasn't clear to me. "In support, most genes peaked in frequency either during 2011 (year with the lowest temperature) or 2015 (year with the highest temperature) (Fig. 4B)." If you are looking at relative abundance data, how could it be that most genes peaked in frequency. I could be missing something about how the z score normalization should be interpreted. Even so, more clarification here would be helpful.

LN183 The same is true here... "genes most frequent during cold.". Frequency is most commonly associated with how often something occurs, like a rate. Here, I take frequency as the relative abundance or proportion of a particular taxon or functional potential within a metagenomic sample. I would suggest defining this upfront, so that it is clear that we are looking at proportional shifts, not absolute.

LN226 and on... Throughout this section on gene enrichment, here again it seems like it would be more accurate to say proportion instead of frequency. For example, instead of "The maximum frequency of Fe stress genes matched periods with high macronutrient supply in part due to upwelling", saying "The maximum proportion of Fe stress genes matched..."

LN236,240,252 Same as above. Suggest replacing frequency with proportion.

LN 256 "P acquisition genes generally followed N genes but reached a maximum in 2018." Suggest adding "maximum proportion".

LN 276. "A time-series from the Mediterranean Sea¹² and now our MiCRO time-series observations suggest parallel taxonomic and functional microbial biodiversity cycles. These observations challenge the notion of functional redundancy in marine microbiomes and instead suggest compositionally as well as functionally very dynamic ecosystems." This statement seems to arise from doing the analysis on anomaly basis, and is potentially erroneous. The functions in the system are more than likely not going away, particularly those carried out by the oligotrophs. If you look across your time series, it is likely that the oligotrophic taxa and their associated functions are not only present, but relatively abundant across most samples. I could be wrong, but the authors need to show the across community proportional data that would either support or refute the statement given their reliance on anomaly analysis.

2. Some analytical methods and processes are not adequately explained within the main text, making it challenging for readers to fully understand the approach. In several places, it would be helpful to the reader to describe the analysis process in slightly more detail. I realize that eventually the methods section gives the details, but while reading through the paper I was consistently wondering basic questions about how the analysis were done.

Detailed Comments:

- LN116: Before delving into metagenomic details, please provide a brief explanation of how metagenomes were analyzed, particularly how read taxonomies were assigned.
- LN133: Briefly explain how genome sizes were estimated (e.g., from reference genomes or MAGs).
- LN140: Provide more context for the MCOA analysis. Explain what MCOA typically aims to achieve and whether your data met the necessary assumptions for this analysis.
- LN141: Clarify what the four functional classification schemes are, as this is the first mention and readers may not be familiar with them.

- Please explicitly state what the anomaly metric is. It is not currently explicitly defined in the methods.
- LN181: Clarify what is meant by "long-term" changes. El Niño events are typically considered short-term; long-term trends would be expected over the entire 11-year dataset.
- LN203-206: The Natural Language Processing model seems interesting, but more detail is needed on its implementation and output, both here and in the methods section.
- LN211: There's a typo with a comma that needs correction.
- LN244: Consider the possibility that organic matter composition might be driving community changes, rather than the other way around.
- LN277: The statement challenging functional redundancy in marine microbiomes is based on relative enrichment and depletion. Consider discussing the limitations of this interpretation based on relative abundance data.

- LN280: Reconsider the use of the term "succession" given that the observations are based on enrichment and depletion rather than absolute changes in community composition.

- LN282: Is it possible that copiotrophic taxa just harbor more iron stress genes than oligotrophic taxa, and may not be actively expressing them? Thus the gene proportions are not accurately reflecting the biogeochemical states?

Figures and Data Presentation:

- Some quantitative statements are difficult to confirm with the given figures. For example, the claim that most genes peaked in frequency in 2011 or 2015 (Fig. 4B) is not easily verifiable from the figure. Consider providing more quantitative plots or summary statistics to support such statements.

Recommendations:

1. Revise the manuscript to clearly distinguish between relative and absolute abundance changes. Use "proportion" instead of "frequency" throughout.
2. Incorporate total cell abundance data, if available, to provide context for relative abundance changes.
3. Enhance the description of analytical methods within the main text.
4. Reconsider the use of terms like "succession" and focus on describing observed patterns of enrichment and depletion.
5. Improve figure clarity to better support quantitative claims.
6. Discuss the limitations of interpreting relative abundance data and anomalies.

Version 1:

Reviewer comments:

Reviewer #1

(Remarks to the Author)

The authors have addressed all of the comments made by myself and the other reviewer, and I think the manuscript is suitable for publication.

Reviewer #2

(Remarks to the Author)

The authors have done a good job in trying to address my comments. They have sufficiently addressed my concerns. This is an excellent manuscript and will be a great value to the field.

REVIEWER COMMENTS

Reviewer #1 (Remarks to the Author):

1. This study uses contemporary seasonal and interannual time series data from the southern California Current (metagenomic, biogeochemical, biological, physical) to provide insights into the effects of a changing ocean on the base of the marine food-web. The authors revealed seasonal cycling of marine microbiome function linked to key microbial functional traits and environmental variability over more than a decade. This allowed identification of elevated nutrient stress and decreased organic carbon cycling through the functioning of the microbiome under El Nino conditions (which resemble predicted future conditions), reflecting a shift towards oligotrophic taxa, traits, and functions. The results are noteworthy, as previous studies have lacked the long-term temporal resolution provided here or have been limited by only focussing on microbial taxonomic diversity, rather than function. However, I feel the use of broad classifications of functional genes (iron, nitrogen, and phosphorous stress genes, and carbon genes), potentially masks some of the complexity and intricacies of the ocean microbiome and the biotic and abiotic interactions that determine ecosystem function. While the nuances of iron, nitrogen, and phosphorous cycling genes are demonstrated in the Extended Data Figure, they are discussed in rather a coarse manner in the main manuscript. For example, on lines 226-233, seasonal patterns in "stress genes" and their links to macronutrient supply are detailed, but no further information is provided as to the nature of the stress or the form of iron, nitrogen, or phosphorous that is inferred to be limiting from the functional genes. This idea is planted on line 211 in regards to previous work on phytoplankton, but not thoroughly explored in my opinion. It is also unclear in the text whether the authors have examined whole pathways associated with "nitrogen stress" for example, and how have they have treated incomplete pathways or poorly annotated pathways.

OUR RESPONSE:

This is an excellent point. To address this, we have added new analyses (e.g., Figure S10) and text. In Fig. S10, we show how the dynamics of individual genes support the dynamics of the trait and overall ecosystem state. As the reviewer correctly pointed out, our rationale was quite brief. So we have added additional descriptions of how these genes are associated with stress and picked for this analysis. This includes a clear reference to Supplementary Table 2. Next, we have added a full discussion paragraph about the rationale, implication and uncertainties associated with picking these genes. Finally, we have added a detailed description of the analysis for picking the genes, the

robustness of annotation, and temporal dynamics of these biogeochemical markers. We hope that these combined efforts add 'depth' to our biogeochemical analysis.

2. A similar comment for the carbon cycling section of the manuscript. I felt it was lacking in detailed information about the types of carbon and potential sources/interactions and seasonality that could be inferred from the identity and phylogeny of specific genes associated with carbon availability and processing.

OUR RESPONSE:

We have address this comment similarly to #1 by adding additional analyses, description and discussion. We expand on the focus on large molecules and in particular carbohydrates, as the degradation pathways are well understood. Similar to above, we hope this adds details to our analysis and addresses the reviewer comment.

3. The authors aimed to "establish a genome-enabled understanding of the climate driven feedbacks..." (lines 70-72), however the metagenomic analyses are gene centric. Functional genes were linked to specific taxonomic ranks, but not genomes of individuals or populations of microorganisms. The large number of samples and the range of coverages of individual taxa provided by the highly seasonal nature of the site, should assist the authors in recovering genomes from these samples?

OUR RESPONSE:

We did not appreciate this nuance and have modified the description of the study to avoid using 'genome-enabled' - which as correctly pointed out, refers to something else. The reviewer is also suggesting that we do a genome-resolved analysis. In this study, we aimed to provide a community-level perspective of the seasonal and long-term changes in biodiversity. A genome-resolved study has a very different focus - i.e., the evolution and selection of specific genotypes. Such analyses will require different questions and provide different insights. Thus, we hope the reviewer and editor accepts that we provide such analyses and findings in another paper.

4. Some of the methodological details are lacking which may limit complete reproducibility. The authors mention some analyses performed in Matlab but do not provide information about the commands used (line 438). They also mention the use of custom hidden Markov models that were implemented for the functional annotations but not what they were targeting (line 493). It is unclear

what cut-offs were used to assign functional gene annotations. The authors used a keyword model to determine seasonal patterns in the frequency of specific functional genes based on their annotations as an exploratory tool – this of course can be flawed if annotations are incorrect/incomplete etc. Were further steps taken to confirm the presence of functional motifs in gene sequences associated with key annotations?

OUR RESPONSE:

We have attempted to add more methodological detail throughout the analysis including for the bioinformatic analyses (part of the JGI pipeline), the matlab commands (mainly simple arithmetic), and the keyword model. Finally, we added a new analysis to test the robustness of the annotation (Supplementary Fig. 10).

5. The Extended Data Figures provide useful information in support of the authors conclusions. In Extended Data Figure 1, it would be helpful to highlight the ENSO cycle on these plots. In Extended Data Figure 6, some indication of what the x and y axis represent on the heatmaps (right hand side of the figure) is needed. In Extended Data Figure 7 and 8, additional descriptions of these figures are missing. It would be useful to highlight that the genes included within each pathway are detailed in the Supplementary Table.

OUR RESPONSE:

These are excellent suggestions and we have completed all the figure edits requested. We have also referenced Supplementary Table 2 in the caption to Figures S8-10. However, it is worth noting that we did not do a pathway analysis as most of the genes are what you can consider 'entry point' functions. In other words, they form the initial steps of 'funneling' Fe, P, N or C into central metabolic pathways. This includes transporters or genes transforming more complex versions of the nutrient into the inorganic form (e.g., alkaline phosphatase or nitrate reductase). We have attempted to describe this nuance when we introduce these results and discuss the interpretation later.

6. Overall, the research is of significance to the field of marine science, and will be of interest to climate scientists, microbiologists, and biological oceanographers, as well as more broadly to researchers with an interest in time-series analyses, ecology and biogeochemistry. The manuscript is well-written and interesting, and the authors have succinctly reported on what is clearly a large and complicated dataset.

OUR RESPONSE:

We appreciate your supportive comments and hope you find the revised manuscript improved.

Reviewer #2 (Remarks to the Author):

This manuscript by Larkin et al. presents a comprehensive and innovative analysis of an important 11-year metagenomic dataset from the Southern California Bight. The study offers valuable insights into the seasonal and interannual dynamics of marine microbial communities and their functional potential, particularly in relation to ENSO cycles. This work has the potential to make a substantial contribution to our understanding of climate-driven changes in marine microbial ecosystems. While the research is well-conducted and the manuscript is well-written, there are some concerns regarding data interpretation and presentation to be addressed.

7. A primary concern with this study is the interpretation of relative abundance data and the use of anomalies as a metric. The authors describe oligotrophic functions as increasing or decreasing in frequency in summer and fall, but this may not accurately reflect absolute abundances. It's possible that these functions maintain a consistent absolute abundance year-round, only appearing "depleted" when copiotrophs increase. This study, like many omics analyses, relies on relative abundance data, which can lead to misleading interpretations. While the authors use terms like increased or depleted frequency in some instances, they often imply actual increases or decreases in abundance, which cannot be inferred from relative data alone.

OUR RESPONSE:

The reviewer raises an excellent point about the importance of relative vs. absolute changes for community structure and functioning. To address this comment, we first add a new analysis comparing the relative frequency of *Prochlorococcus* and *Synechococcus* with flow cytometry counts (Supplementary Fig. 2). It is worth noting that we only have three years of count data (2012 -2015). However, we observe significant correlations between proportions and absolute abundances with correlation coefficients $R \sim 0.8$. Occasionally, *Prochlorococcus* was absent in the absolute counts and during these periods the proportion was also near zero - suggesting that low frequencies are indicative of being largely absent from the community. Next, we did a literature review reported in the discussion. Coastal time-series with similar environmental dynamics have reported very limited annual swings in total bacterial counts. This explains the significant correspondence between absolute and relative

quantifications of lineages. It is worth noting that other places with stronger seasonal environmental dynamics result in clear changes in bacterial abundance. Thus, the correspondence seen in our time-series may not be applicable to other locations. We have modified the discussion to address the issue of relative vs. absolute abundances and why they correspond in our case. These issues are now presented in the discussion.

8. An alternative interpretation could be that oligotrophic taxa maintain stable abundances throughout the year, while copiotrophic taxa increase during spring's heightened primary production. This scenario would show the same relative abundance patterns but with very different ecological implications. The use of the term "succession" is particularly problematic, as it implies species displacement over time as niches evolve. It's unclear if this accurately describes the dynamics in the Southern California Current. Are copiotrophs truly displacing oligotrophs, or are they simply increasing in abundance alongside a stable oligotrophic population?

OUR RESPONSE:

We do not see support for a stable 'background' abundance of oligotrophs. For example, *Prochlorococcus* is commonly regarded as a model bacterium for oligotrophic proliferation, and both the abundance and frequency of this lineage are dynamic at our site. This is described with our new figure (Supplementary Figure 2). We also present these two alternative models in the discussion and how we interpret our data to support a dynamic community.

The reviewer also raises the issue of how we use the term 'succession'. We recognize that there are multiple definitions of this term from terrestrial and aquatic ecology. These theories include whether it implies displacement or simply changes in dominance. In fire ecology, the former version is often implied. However, Hutchinson, Margalef, Reynolds, etc. use the latter version to describe seasonal changes in plankton communities and niche evolution. To avoid confusion, we now cite 'Reynolds' so readers can identify the source of our definition of this keyword.

9. To address these limitations, the authors should discuss the constraints of relative abundance analysis and using anomaly as a metric. Additionally, incorporating total cell abundance data, if available, could provide crucial context. This would help distinguish between actual changes in community composition and increases in copiotrophic taxa overlaying a consistent oligotrophic

population. If such data were collected, normalizing to total abundances could reveal which taxa and functions remain constant throughout the year and which exhibit true dynamics.

OUR RESPONSE:

We hope the combination of our new analyses and discussion address this important point. Overall, we interpret the observations as dynamic changes for all the abundant members.

Here are several lines that exemplify this:

Ln 69 - “Thus, we predict that El Niño-driven warming will result in increases of oligotrophic bacterioplankton taxa with a concurrent increase in metabolic strategies associated with oligotrophic communities.”

Ln 122 “These common taxa showed clear seasonal succession (Fig. 2B).”

LN 129 “In the fall Pelagibacteraceae was common...”. This implies that Pelagibacteraceae was not common in the other seasons, which seems highly unlikely. How often in the dataset are the Pelagibacteracea in the top 3 most abundant families, if not the very top. This is often the case in most marine systems.

OUR RESPONSE:

We agree with the reviewer that this was poor wording. We have changed it to ‘more common’ as we never meant to imply that Pelagibacteraceae was anything but common.

LN 178 This statement wasn’t clear to me. “In support, most genes peaked in frequency either during 2011 (year with the lowest temperature) or 2015 (year with the highest temperature) (Fig. 4B).” If you are looking at relative abundance data, how could it be that most genes peaked in frequency. I could be missing something about how the z score normalization should be interpreted. Even so, more clarification here would be helpful.

OUR RESPONSE:

You did not misunderstand our analysis, and we agree that this was a poorly worded statement. So we have removed it altogether.

LN183 The same is true here... “genes most frequent during cold,”. Frequency is most commonly associated with how often something occurs, like a rate. Here, I take frequency as the relative abundance or proportion of a particular taxon or functional

potential within a metagenomic sample. I would suggest defining this upfront, so that it is clear that we are looking at proportional shifts, not absolute.

OUR RESPONSE:

This is an issue of semantics and to address it, we have taken your suggestion and replaced frequency with proportion throughout the manuscript.

LN226 and on... Throughout this section on gene enrichment, here again it seems like it would be more accurate to say proportion instead of frequency. For example, instead of “The maximum frequency of Fe stress genes matched periods with high macronutrient supply in part due to upwelling”, saying “The maximum proportion of Fe stress genes matched...”

OUR RESPONSE:

We agree and have replaced it with proportion (as in the comment above).

LN236,240,252 Same as above. Suggest replacing frequency with proportion.

OUR RESPONSE:

We agree and have replaced it with proportion (as in the comment above).

LN 256 “P acquisition genes generally followed N genes but reached a maximum in 2018.” Suggest adding “maximum proportion”.

OUR RESPONSE:

We agree and have replaced it with proportion (as in the comment above).

LN 276. “A time-series from the Mediterranean Sea and now our MiCRO time-series observations suggest parallel taxonomic and functional microbial biodiversity cycles. These observations challenge the notion of functional redundancy in marine microbiomes and instead suggest compositionally as well as functionally very dynamic ecosystems.” This statement seems to arise from doing the analysis on anomaly basis, and is potentially erroneous. The functions in the system are more than likely not going away, particularly those carried out by the oligotrophs. If you look across your time series, it is likely that the oligotrophic taxa and their associated functions are not only present, but relatively abundant across most samples. I could be wrong, but the authors need to show the across community proportional data that would either support or refute the statement given their reliance on anomaly analysis.

OUR RESPONSE:

We hope that the comparison between absolute counts and relative proportions for *Prochlorococcus* and *Synechococcus* as well as our added literature review of changes in bacterial counts in other systems address this comment. From these comparisons, we argue that there is evidence of substantial oscillations in many lineages and

functions - including ones encoding for nutrient stress that we would normally associate with oligotrophic conditions.

10. Some analytical methods and processes are not adequately explained within the main text, making it challenging for readers to fully understand the approach. In several places, it would be helpful to the reader to describe the analysis process in slightly more detail. I realize that eventually the methods section gives the details, but while reading through the paper I was consistently wondering basic questions about how the analysis were done.

OUR RESPONSE:

This is an issue related to the fact that many of the method details are at the end. This is a journal choice that we do not have any influence on. However, we agree that the issue can be frustrating. To partially address it, we have added additional methodological details in the result section to indicate the analysis being employed. We hope that this makes the analysis and results easier to follow. We are trying to strike the balance between focusing on the results while providing just enough methodological detail upfront for the reader to follow. We are happy to adjust this if needed.

Detailed Comments:

- LN116: Before delving into metagenomic details, please provide a brief explanation of how metagenomes were analyzed, particularly how read taxonomies were assigned.

OUR RESPONSE:

This has now been added.

- LN133: Briefly explain how genome sizes were estimated (e.g., from reference genomes or MAGs).

OUR RESPONSE:

This is now briefly described. Furthermore, we have expanded the description in the methods.

- LN140: Provide more context for the MCOA analysis. Explain what MCOA typically aims to achieve and whether your data met the necessary assumptions for this analysis.

OUR RESPONSE:

We have added this methodological detail.

- LN141: Clarify what the four functional classification schemes are, as this is the first mention and readers may not be familiar with them.

OUR RESPONSE:

This information has been added

- Please explicitly state what the anomaly metric is. It is not currently explicitly defined in the methods.

OUR RESPONSE:

We have modified the methods to directly define anomaly (estimated from fitting the linear models). We also describe this approach in the caption to Figure 1.

- LN181: Clarify what is meant by "long-term" changes. El Niño events are typically considered short-term; long-term trends would be expected over the entire 11-year dataset.

OUR RESPONSE:

We have modified this to interannual to avoid any confusion.

- LN203-206: The Natural Language Processing model seems interesting, but more detail is needed on its implementation and output, both here and in the methods section.

OUR RESPONSE:

We added additional text to clarify the use. It is important to note that we never present any quantitative outputs from this analysis and only use it for exploration as annotation keywords are not the same as biodiversity. Any quantitative analyses are presented in Figures 5, Supplementary 8-10.

- LN211: There's a typo with a comma that needs correction.

OUR RESPONSE:

Corrected

- LN244: Consider the possibility that organic matter composition might be driving community changes, rather than the other way around.

OUR RESPONSE:

We completely agree and have modified the discussion to reflect this viewpoint.

- LN277: The statement challenging functional redundancy in marine microbiomes is based on relative enrichment and depletion. Consider discussing the limitations of this interpretation based on relative abundance data.

OUR RESPONSE:

This point follows earlier comments. We have modified this paragraph to discuss the two different models and why we think observations support this succession model.

- LN280: Reconsider the use of the term "succession" given that the observations are based on enrichment and depletion rather than absolute changes in community composition.

OUR RESPONSE:

This is similar to an earlier comment. We recognize that there are multiple definitions of this term from terrestrial and aquatic ecology. These theories include whether it implies displacement or simply changes in dominance. In fire ecology, the former version is often implied. However, Hutchinson, Margalef, Reynolds, etc. use the latter version to describe seasonal changes in plankton communities and niche evolution. To avoid confusion, we now cite 'Reynolds; so readers can identify the source of our definition of this keyword.

-LN282: Is it possible that copiotrophic taxa just harbor more iron stress genes than oligotrophic taxa, and may not be actively expressing them? Thus the gene proportions are not accurately reflecting the biogeochemical states?

OUR RESPONSE:

This is an excellent point that is hard to completely address. In the revised version, we now discussed this issue. In general, we interpret our data as reflective of adaptation to the wider ocean environment. We also reference this comment and state that metatranscriptomics is required if you want instantaneous responses to environmental changes.

Figures and Data Presentation:

- Some quantitative statements are difficult to confirm with the given figures. For example, the claim that most genes peaked in frequency in 2011 or 2015 (Fig. 4B) is not easily verifiable from the figure. Consider providing more quantitative plots or summary statistics to support such statements.

OUR RESPONSE:

This is an excellent point. The other reviewer called out the peak frequency and we have removed this statement. Furthermore, we have attempted to clarify all quantitative statements to ensure transparency.

Recommendations:

1. Revise the manuscript to clearly distinguish between relative and absolute abundance changes. Use "proportion" instead of "frequency" throughout.
2. Incorporate total cell abundance data, if available, to provide context for relative abundance changes.
3. Enhance the description of analytical methods within the main text.

4. Reconsider the use of terms like "succession" and focus on describing observed patterns of enrichment and depletion.
5. Improve figure clarity to better support quantitative claims.
6. Discuss the limitations of interpreting relative abundance data and anomalies.

OUR RESPONSE:

We have addressed all the issues and hope you find the manuscript improved.